# Solving the multi-way matching problem by permutation synchronization

**Deepti Pachauri,**[†] **Risi Kondor**[§] **and Vikas Singh**[‡†]
[†]Dept. of Computer Sciences, University of Wisconsin–Madison
[‡]Dept. of Biostatistics & Medical Informatics, University of Wisconsin–Madison
[§]Dept. of Computer Science and Dept. of Statistics, The University of Chicago
pachauri@cs.wisc.edu   risi@uchicago.edu   vsingh@biostat.wisc.edu

## Abstract

The problem of matching not just two, but $m$ different sets of objects to each other arises in many contexts, including finding the correspondence between feature points across multiple images in computer vision. At present it is usually solved by matching the sets pairwise, in series. In contrast, we propose a new method, Permutation Synchronization, which finds all the matchings jointly, in one shot, via a relaxation to eigenvector decomposition. The resulting algorithm is both computationally efficient, and, as we demonstrate with theoretical arguments as well as experimental results, much more stable to noise than previous methods.

## 1   Introduction

Finding the correct bijection between two sets of objects $X = \{x_1, x_2, \ldots, x_n\}$ and $X' = \{x'_1, x'_2, \ldots, x'_n\}$ is a fundametal problem in computer science, arising in a wide range of contexts [1]. In this paper, we consider its generalization to matching not just two, but $m$ different sets $X_1, X_2, \ldots, X_m$. Our primary motivation and running example is the classic problem of matching landmarks (feature points) across many images of the same object in computer vision, which is a key ingredient of image registration [2], recognition [3, 4], stereo [5], shape matching [6, 7], and structure from motion (SFM) [8, 9]. However, our approach is fully general and equally applicable to problems such as matching multiple graphs [10, 11].

Presently, multi-matching is usually solved sequentially, by first finding a putative permutation $\tau_{12}$ matching $X_1$ to $X_2$, then a permutation $\tau_{23}$ matching $X_2$ to $X_3$, and so on, up to $\tau_{m-1,m}$. While one can conceive of various strategies for optimizing this process, the fact remains that when the data are noisy, a single error in the sequence will typically create a large number of erroneous pairwise matches [12, 13, 14]. In contrast, in this paper we describe a new method, Permutation Synchronization, that estimates the entire matrix $(\tau_{ji})_{i,j=1}^{m}$ of assignments *jointly*, in a single shot, and is therefore much more robust to noise.

For consistency, the recovered matchings must satisfy $\tau_{kj}\tau_{ji} = \tau_{ki}$. While finding an optimal matrix of permutations satisfying these relations is, in general, combinatorially hard, we show that for the most natural choice of loss function the problem has a natural relaxation to just finding the $n$ leading eigenvectors of the cost matrix. In addition to vastly reducing the computational cost, using recent results from random matrix theory, we show that the eigenvectors are very effective at aggregating information from all $\binom{m}{2}$ pairwise matches, and therefore make the algorithm surprisingly robust to noise. Our experiments show that in landmark matching problems Permutation Synchronization can recover the correct correspondence between landmarks across a large number of images with small error, even when a significant fraction of the pairwise matches are incorrect.

The term "synchronization" is inspired by the recent celebrated work of Singer et al. on a similar problem involving finding the right rotations (rather than matchings) between electron microscopic

images [15][16][17]. Historically, multi-matching has received relatively little attention. However, independently of, and concurrently with the present work, Huang and Guibas [18] have recently proposed a semidefinite programming based solution, which parallels our approach, and in problems involving occlusion might perform even better.

## 2  Synchronizing permutations

Consider a collection of $m$ sets $X_1, X_2, \ldots, X_m$ of $n$ objects each, $X_i = \{x_1^i, x_2^i, \ldots, x_n^i\}$, such that for each pair $(X_i, X_j)$, each $x_p^i$ in $X_i$ has a natural counterpart $x_q^j$ in $X_j$. For example, in computer vision, given $m$ images of the same scene taken from different viewpoints, $x_1^i, x_2^i, \ldots, x_n^i$ might be $n$ visual landmarks detected in image $i$, while $x_1^j, x_2^j, \ldots, x_n^j$ are $n$ landmarks detected in image $j$, in which case $x_p^i \sim x_q^j$ signifies that $x_p^i$ and $x_q^j$ correspond to the same physical feature.

Since the correspondence between $X_i$ and $X_j$ is a bijection, one can write it as $x_p^i \sim x_{\tau_{ji}(p)}^j$ for some permutation $\tau_{ji} \colon \{1, 2, \ldots, n\} \to \{1, 2, \ldots, n\}$. Key to our approach to solving multi-matching is that with respect to the natural definition of multiplication, $(\tau'\tau)(i) := (\tau'(\tau(i)))$, the $n!$ possible permutations of $\{1, 2, \ldots, n\}$ form a *group*, called the symmetric group of degree $n$, denoted $\mathbb{S}_n$.

We say that the system of correspondences between $X_1, X_2, \ldots, X_m$ is *consistent* if $x_p^i \sim x_q^j$ and $x_q^j \sim x_r^k$ together imply that $x_p^i \sim x_r^k$. In terms of permutations this is equivalent to requiring that the array $(\tau_{ij})_{i,j=1}^m$ satisfy

$$\tau_{kj}\tau_{ji} = \tau_{ki} \qquad \forall i, j, k. \tag{1}$$

Alternatively, given some reference ordering of $x_1, x_2, \ldots, x_n$, we can think of each $X_i$ as realizing its own permutation $\sigma_i$ (in the sense of $x_\ell \sim x_{\sigma_i(\ell)}^i$), and then $\tau_{ji}$ becomes

$$\tau_{ji} = \sigma_j \sigma_i^{-1}. \tag{2}$$

The existence of permutations $\sigma_1, \sigma_2, \ldots, \sigma_m$ satisfying (2) is equivalent to requiring that $(\tau_{ji})_{i,j=1}^m$ satisfy (1). Thus, assuming consistency, solving the multi-matching problem reduces to finding just $m$ different permutations, rather than $O(m^2)$. However, the $\sigma_i$'s are of course not directly observable. Rather, in a typical application we have some tentative (noisy) $\tilde{\tau}_{ji}$ matchings which we must *synchronize* into the form (2) by finding the underlying $\sigma_1, \ldots, \sigma_m$.

Given $(\tilde{\tau}_{ji})_{i,j=1}^m$ and some appropriate distance metric $d$ between permutations, we formalize Permutation Synchronization as the combinatorial optimization problem

$$\underset{\sigma_1, \sigma_2, \ldots, \sigma_m \in \mathbb{S}_n}{\text{minimize}} \sum_{i,j=1}^N d(\sigma_j \sigma_i^{-1}, \tilde{\tau}_{ji}). \tag{3}$$

The computational cost of solving (3) depends critically on the form of the distance metric $d$. In this paper we limit ourselves to the simplest choice

$$d(\sigma, \tau) = n - \langle P(\sigma), P(\tau) \rangle, \tag{4}$$

where $P(\sigma) \in \mathbb{R}^{n \times n}$ are the usual permutation matrices

$$[P(\sigma)]_{q,p} := \begin{cases} 1 & \text{if } \sigma(p) = q \\ 0 & \text{otherwise,} \end{cases}$$

and $\langle A, B \rangle$ is the matrix inner product $\langle A, B \rangle := \text{tr}(A^\top B) = \sum_{p,q=1}^n A_{p,q} B_{p,q}$.

The distance (4) simply counts the number of objects assigned *differently* by $\sigma$ and $\tau$. Furthermore, it allows us to rewrite (3) as $\text{maximize}_{\sigma_1, \sigma_2, \ldots, \sigma_m} \sum_{i,j=1}^m \langle P(\sigma_j \sigma_i^{-1}), P(\tilde{\tau}_{ji}) \rangle$, suggesting the generalization

$$\underset{\sigma_1, \sigma_2, \ldots, \sigma_m}{\text{maximize}} \sum_{i,j=1}^m \langle P(\sigma_j \sigma_i^{-1}), T_{ji} \rangle, \tag{5}$$

where the $T_{ji}$'s can now be any matrices, subject to $T_{ji}^\top = T_{ij}$. Intuitively, each $T_{ji}$ is an objective matrix, the $(q, p)$ element of which captures the utility of matching $x_p^i$ in $X_i$ to $x_q^j$ in $X_j$. This generalization is very useful when the assignments of the different $x_p^i$'s have different confidences. For example, in the landmark matching case, if, due to occlusion or for some other reason, the counterpart of $x_p^i$ is not present in $X_j$, then we can simply set $[T_{ji}]_{q,p} = 0$ for all $q$.

## 2.1 Representations and eigenvectors

The generalized Permutation Synchronization problem (5) can also be written as

$$\underset{\sigma_1,\sigma_2,\ldots,\sigma_m}{\text{maximize}} \langle \mathcal{P}, \mathcal{T} \rangle, \tag{6}$$

where

$$\mathcal{P} = \begin{pmatrix} P(\sigma_1 \sigma_1^{-1}) & \ldots & P(\sigma_1 \sigma_m^{-1}) \\ \vdots & \ddots & \vdots \\ P(\sigma_m \sigma_1^{-1}) & \ldots & P(\sigma_m \sigma_m^{-1}) \end{pmatrix} \quad \text{and} \quad \mathcal{T} = \begin{pmatrix} T_{11} & \ldots & T_{1m} \\ \vdots & \ddots & \vdots \\ T_{m1} & \ldots & T_{mm} \end{pmatrix}. \tag{7}$$

A matrix valued function $\rho \colon \mathbb{S}_n \to \mathbb{C}^{d \times d}$ is said to be a representation of the symmetric group if $\rho(\sigma_2)\,\rho(\sigma_1) = \rho(\sigma_2\sigma_1)$ for any pair of permutations $\sigma_1, \sigma_2 \in \mathbb{S}_n$. Clearly, $P$ is a representation of $\mathbb{S}_n$ (actually, the so-called defining representation), since $P(\sigma_2\sigma_1) = P(\sigma_2)\,P(\sigma_1)$. Moreover, $P$ is a so-called orthogonal representation, because each $P(\sigma)$ is real and $P(\sigma^{-1}) = P(\sigma)^\top$. Our fundamental observation is that this implies that $\mathcal{P}$ has a very special form.

**Proposition 1.** *The synchronization matrix $\mathcal{P}$ is of rank $n$ and is of the form $\mathcal{P} = U \cdot U^\top$, where*

$$U = \begin{pmatrix} P(\sigma_1) \\ \vdots \\ P(\sigma_m) \end{pmatrix}.$$

**Proof.** From $P$ being a representation of $\mathbb{S}_n$,

$$\mathcal{P} = \begin{pmatrix} P(\sigma_1)\,P(\sigma_1)^\top & \ldots & P(\sigma_1)\,P(\sigma_m)^\top \\ \vdots & \ddots & \vdots \\ P(\sigma_m)\,P(\sigma_1)^\top & \ldots & P(\sigma_m)\,P(\sigma_m)^\top \end{pmatrix}, \tag{8}$$

implying $\mathcal{P} = U \cdot U^\top$. Since $U$ has $n$ columns, $\text{rank}(\mathcal{P})$ is at most $n$. This rank is achieved because $P(\sigma_1)$ is an orthogonal matrix, therefore it has linearly independent columns, and consequently the columns of $U$ cannot be linearly dependent. ∎

**Corollary 1.** *Letting $[P(\sigma_i)]_p$ denote the $p$'th column of $P(\sigma_i)$, the normalized columns of $U$,*

$$u_\ell = \frac{1}{\sqrt{m}} \begin{pmatrix} [P(\sigma_1)]_\ell \\ \vdots \\ [P(\sigma_m)]_\ell \end{pmatrix} \qquad \ell = 1,\ldots,n, \tag{9}$$

*are mutually orthogonal unit eigenvectors of $\mathcal{P}$ with the same eigenvalue $m$, and together span the row/column space of $\mathcal{P}$.*

**Proof.** The columns of $U$ are orthogonal because the columns of each constituent $P(\sigma_i)$ are orthogonal. The normalization follows from each column of $P(\sigma_i)$ having norm 1. The rest follows by Proposition 1. ∎

## 2.2 An easy relaxation

Solving (6) is computationally difficult, because it involves searching the combinatorial space of a combination of $m$ permutations. However, Proposition 1 and its corollary suggest relaxing it to

$$\underset{\mathcal{P} \in \mathfrak{M}_n^m}{\text{maximize}} \langle \mathcal{P}, \mathcal{T} \rangle, \tag{10}$$

where $\mathfrak{M}_n^m$ is the set of $mn$–dimensional rank $n$ symmetric matrices whose non-zero eigenvalues are $m$. This is now just a generalized Rayleigh problem, the solution of which is simply

$$\mathcal{P} = m \sum_{\ell=1}^{n} v_\ell\, v_\ell^\top, \tag{11}$$

where $v_1, v_2, \ldots, v_\ell$ are the $n$ leading normalized eigenvectors of $\mathcal{T}$. Equivalently, $\mathcal{P} = U \cdot U^\top$, where

$$U = \sqrt{m} \begin{pmatrix} | & | & \cdots & | \\ v_1 & v_2 & \cdots & v_n \\ | & | & \cdots & | \end{pmatrix}. \tag{12}$$

Thus, in contrast to the original combinatorial problem, (10) can be solved by just finding the $m$ leading eigenvectors of $\mathcal{T}$.

Of course, from $\mathcal{P}$ we must still recover the individual permutations $\sigma_1, \sigma_2, \ldots, \sigma_m$. However, as long as $\mathcal{P}$ is relatively close in form (7), this is quite a simple and stable process. One way to do it is to let each $\sigma_i$ be the permutation that best matches the $(i, 1)$ block of $\mathcal{P}$ in the linear assignment sense,

$$\sigma_i = \arg \min_{\sigma \in \mathbb{S}_n} \langle P(\sigma), [\mathcal{P}]_{i,1} \rangle,$$

which is solved in $O(n^3)$ time by the Kuhn–Munkres algorithm [19][1], and then set $\tau_{ji} = \sigma_j \sigma_i^{-1}$, which will then satisfy the consistency relations. The pseudocode of the full algorithm is given in Algorithm 1.

---

**Algorithm 1** Permutation Synchronization

**Input:** the objective matrix $\mathcal{T}$
  **Compute** the $n$ leading eigenvectors $(v_1, v_2, \ldots, v_n)$ of $\mathcal{T}$ and set $U = \sqrt{m}\,[v_1, v_2, \ldots, v_n]$
  **for** $i = 1$ to $m$ **do**
    $P_{i1} = U_{(i-1)n+1:in,\,1:n}\, U_{1:n,\,1:n}^\top$
    $\sigma_i = \arg \max_{\sigma \in \mathbb{S}_n} \langle P_{i1}, \sigma \rangle$   [Kuhn-Munkres]
  **end for**
  **for** each $(i, j)$ **do**
    $\tau_{ji} = \sigma_j \sigma_i^{-1}$
  **end for**
**Output:** the matrix $(\tau_{ji})_{i,j=1}^m$ of globally consistent matchings

---

## 3 Analysis of the relaxed algorithm

Let us now investigate under what conditions we can expect the relaxation (10) to work well, in particular, in what cases we can expect the recovered matchings to be *exact*.

In the absence of noise, i.e., when $T_{ji} = P(\tilde{\tau}_{ji})$ for some array $(\tilde{\tau}_{ji})_{j,i}$ of permutations that already satisfy the consistency relations (1), $\mathcal{T}$ will have precisely the same structure as described by Proposition 1 for $\mathcal{P}$. In particular, it will have $n$ mutually orthogonal eigenvectors

$$v_\ell = \frac{1}{\sqrt{m}} \begin{pmatrix} [P(\tilde{\sigma}_1)]_\ell \\ \vdots \\ [P(\tilde{\sigma}_m)]_\ell \end{pmatrix} \qquad \ell = 1, \ldots, n \tag{13}$$

with the same eigenvalue $m$. Due to the $n$–fold degeneracy, however, the matrix of eigenvectors (12) is only defined up to multiplication by an arbitrary rotation matrix $O$ on the right, which means that instead of the "correct" $U$ (whose columns are (13)), the eigenvector decomposition of $\mathcal{T}$ may return any $U' = UO$. Fortunately, when forming the product

$$\mathcal{P} = U' \cdot U'^\top = U\, O\, O^\top U^\top = U \cdot U^\top$$

this rotation cancels, confirming that our algorithm recovers $\mathcal{P} = \mathcal{T}$, and hence the matchings $\tau_{ji} = \tilde{\tau}_{ji}$, with no error.

Of course, rather than the case when the solution is handed to us from the start, we are more interested in how the algorithm performs in situations when either the $T_{ji}$ blocks are not permutation matrices, or they are not synchronized. To this end, we set

$$\mathcal{T} = \mathcal{T}_0 + \mathcal{N}, \tag{14}$$

where $\mathcal{T}_0$ is the correct "ground truth" synchronization matrix, while $\mathcal{N}$ is a symmetric perturbation matrix with entries drawn independently from a zero-mean normal distribution with variance $\eta^2$.

In general, to find the permutation best aligned with a given $n \times n$ matrix $T$, the Kuhn–Munkres algorithm solves for $\hat{\tau} = \arg \max_{\tau \in \mathbb{S}_n} \langle P(\tau), T \rangle = \arg \max_{\tau \in \mathbb{S}_n} (\text{vec}(P(\tau)) \cdot \text{vec}(T))$. Therefore,

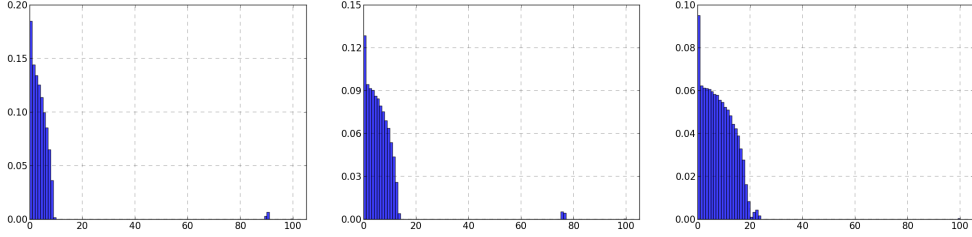

Figure 1: Singular value histogram of $\mathcal{T}$ under the noise model where each $\tilde{\tau}_{ji}$ with probability $p = \{0.10, 0.25, 0.85\}$ is replaced by a random permutation ($m = 100$, $n = 30$). Note that apart from the extra peak at zero, the distribution of the stochastic eigenvalues is very similar to the semicircular distribution for Gaussian noise. As long as the small cluster of deterministic eigenvalues is clearly separated from the noise, Permutation Synchronization is feasible.

writing $T = P(\tau_0) + \epsilon$, where $P(\tau_0)$ is the "ground truth", while $\epsilon$ is an error term, it is guaranteed to return the correct permutation as long as

$$\| \operatorname{vec}(\epsilon) \| < \min_{\tau' \in \mathbb{S}_n \setminus \{\tau_0\}} \| \operatorname{vec}(\tau_0) - \operatorname{vec}(\tau') \| / 2.$$

By the symmetry of $\mathbb{S}_n$, the right hand side is the same for any $\tau_0$, so w.l.o.g. we can set $\tau_0 = e$ (the identity), and find that the minimum is achieved when $\tau'$ is just a transposition, e.g., the permutation that swaps $1$ with $2$ and leaves $3, 4, \ldots, n$ in place. The corresponding permutation matrix differs from the idenity in exactly $4$ entries, therefore a sufficient condition for correct reconstruction is that $\|\epsilon\|_{\mathrm{Frob}} = \langle \epsilon, \epsilon \rangle^{1/2} = \|\operatorname{vec}(\epsilon)\| < \frac{1}{2}\sqrt{4} = 1$. As $n$ grows, $\|\epsilon\|_{\mathrm{Frob}}$ becomes tightly concentrated around $\eta n$, so the condition for recovering the correct permutation is $\eta < 1/n$.

Permutation Synchronization can achieve a lower error, especially in the large $m$ regime, because the eigenvectors aggregate information from all the $T_{ji}$ matrices, and tend to be very stable to perturbations. In general, perturbations of the form (14) exhibit a characteristic phase transition. As long as the largest eigenvalue of the random matrix $\mathcal{N}$ falls below a given multiple of the smallest non-zero eigenvalue of $\mathcal{T}_0$, adding $\mathcal{N}$ will have very little effect on the eigenvectors of $\mathcal{T}$. On the other hand, when the noise exceeds this limit, the spectra get fully mixed, and it becomes impossible to recover $\mathcal{T}_0$ from $\mathcal{T}$ to any precision at all.

If $\mathcal{N}$ is a symmetric matrix with independent $\mathcal{N}(0, \eta^2)$ entries, as $nm \to \infty$, its spectrum will tend to Wigner's famous semicircle distribution supported on the interval $(-2\eta(nm)^{1/2}, 2\eta(nm)^{1/2})$, and with probability one the largest eigenvalue will approach $2\eta(nm)^{1/2}$ [20, 21]. In contrast, the non-zero eigenvalues of $\mathcal{T}_0$ scale with $m$, which guarantees that for large enough $m$ the two spectra will be nicely separated and Permutation Synchronization will have very low error. While much harder to analyze analytically, empirical evidence suggests that this type of phase transition behavior is characteristic of any reasonable noise model, for example the one in which we take each block of $\mathcal{T}$ and with some probability $p$ replace it with a random permutation matrix (Figure 1).

To derive more quantitative results, we consider the case where $\mathcal{N}$ is a so-called (symmetric) Gaussian Wigner matrix, which has independent $\mathcal{N}(0, \eta^2)$ entries on its diagonal, and $\mathcal{N}(0, \eta^2/2)$ entries everywhere else. It has recently been proved that for this type of matrix the phase transition occurs at $\lambda_{\mathrm{min}}^{\mathrm{det}} / \lambda_{\mathrm{max}}^{\mathrm{stochastic}} = 1/2$, so to recover $\mathcal{T}_0$ to any accuracy at all we must have $\eta < (m/n)^{1/2}$ [22]. Below this limit, to quantify the actual expected error, we write each leading normalized eigenvector $v_1, v_2, \ldots, v_n$ of $\mathcal{T}$ as $v_i = v_i^* + v_i^\perp$, where $v_i^*$ is the projection of $v_i$ to the space $\mathcal{U}_0$ spanned by the non-zero eigenvectors $v_1^0, v_2^0, \ldots, v_n^0$ of $\mathcal{T}_0$. By Theorem 2.2 of [22] as $nm \to \infty$,

$$\|v_i^*\|^2 \xrightarrow{a.s.} 1 - \eta^2 \frac{n}{m} \qquad \text{and} \qquad \|v_i^\perp\|^2 \xrightarrow{a.s.} \eta^2 \frac{n}{m} \,. \tag{15}$$

It is easy to see that $\langle v_i^\perp, v_j^\perp \rangle \xrightarrow{a.s.} 0$, which implies $\langle v_i^*, v_j^* \rangle = \langle v_i, v_j \rangle - \langle v_i^\perp, v_j^\perp \rangle \xrightarrow{a.s.} 0$, so, setting $\lambda = (1 - \eta^2 n/m)^{-1/2}$, the normalized vectors $\lambda v_1^*, \ldots, \lambda v_n^*$ almost surely tend to an orthonormal basis for $\mathcal{U}_0$. Thus, $U = \sqrt{m}\,[v_1, \ldots, v_n]$ is related to the "true" $U_0 = \sqrt{m}\,[v_1^0, \ldots, v_n^0]$ by

$$\lambda U \xrightarrow{a.s.} U_0 O + \lambda E' = (U_0 + \lambda E)O,$$

where $O$ is some rotation and each column of the noise matrices $E$ and $E'$ has norm $\eta(n/m)^{1/2}$. Since multiplying $U$ on the right by an orthogonal matrix does not affect $\mathcal{P}$, and the Kuhn–Munkres

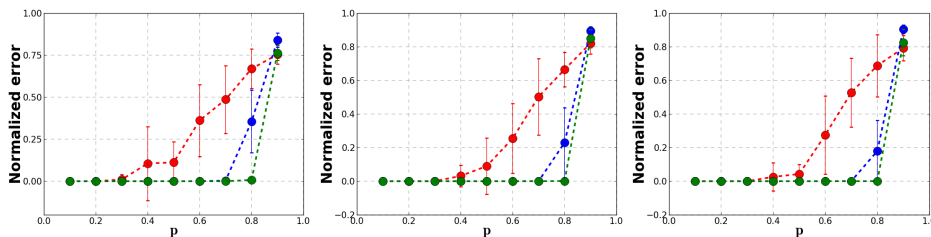

Figure 2: The fraction of $(\sigma_i)_{i=1}^m$ permutations that are incorrect when reconstructed by Permutation Synchronization from an array $(\tilde{\tau}_{ji})_{j,i=1}^m$, in which each entry, with probability $p$ is replaced by a random permutation. The plots show the mean and standard deviation of errors over 20 runs as a function of $p$ for $m=10$ (red), $m=50$ (blue) and $m=100$ (green). (Left) $n=10$. (Center) $n=25$. (Right) $n=30$.

algorithm is invariant to scaling by a constant, this equation tells us that (almost surely) the effect of (14) is equivalent to setting $U = U_0 + \lambda E$. In terms of the individual $P_{ji}$ blocks of $\mathcal{P} = UU^\top$, neglecting second order terms,

$$P_{ji} = (U_j^0 + \lambda E_j)(U_i^0 + \lambda E_i)^\top \approx P(\tau_{ji}) + \lambda U_j^0 E_i^\top + \lambda E_j U_i^{0\top},$$

where $\tau_{ji}$ is the ground truth matching and $U_i^0$ and $E_i$ denote the appropriate $n \times n$ submatrices of $U^0$ and $E$. Conjecturing that in the limit $E_i$ and $E_j$ follow rotationally invariant distributions, almost surely

$$\lim \| U_j^0 E_i^\top + E_j U_i^{0\top} \|_{\text{Frob}} = \lim \| E_i + E_j \|_{\text{Frob}} \leq 2\eta n/m.$$

Thus, plugging in to our earlier result for the error tolerance of the Kuhn–Munkres algorithm, Permutation Synchronization will correctly recover $\tau_{ji}$ with probability one provided $2\lambda \eta n/m < 1$, or, equivalently,

$$\eta^2 < \frac{m/n}{1 + 4(m/n)^{-1}} .$$

This is much better than our $\eta < 1/n$ result for the naive algorithm, and remarkably only slightly stricter than the condition $\eta < (m/n)^{1/2}$ for recovering the eigenvectors with any accuracy at all. Of course, these results are asymptotic (in the sense of $nm \to \infty$), and strictly speaking only apply to additive Gaussian Wigner noise. However, as Figure 2 shows, in practice, even when the noise is in the form of corrupting entire permutations and $nm$ is relatively small, qualitatively our algorithm exhibits the correct behavior, and for large enough $m$ Permutation Synchronization does indeed recover *all* $(\tau_{ji})_{j,i=1}^m$ with no error even when the vast majority of the entries in $\mathcal{T}$ are incorrect.

## 4 Experiments

Since computer vision is one of the areas where improving the accuracy of multi-matching problems is the most pressing, our experiments focused on this domain. For a more details of our results, please see the extended version of the paper available on project website.

**Stereo Matching.** As a proof of principle, we considered the task of aligning landmarks in 2D images of the same object taken from different viewpoints in the **CMU house** ($m = 111$ frames of a video sequence of a toy house with $n = 30$ hand labeled landmark points in each frame) and **CMU hotel** ($m = 101$ frames of a video sequence of a toy hotel, $n = 30$ hand labeled landmark points in each frame) datasets. The baseline method is to compute $(\tilde{\tau}_{ji})_{i,j=1}^m$ by solving $\binom{m}{2}$ independent linear assignment problems based on matching landmarks by their shape context features [23]. Our method takes the same pairwise matches and synchronizes them with the eigenvector based procedure. Figure 3 shows that this clearly outperforms the baseline, which tends to degrade progressively as the number of images increases. This is due to the fact that the appearance (or descriptors) of keypoints differ considerably for large offset pairs (which is likely when the image set is large), leading to many false matches. In contrast, our method improves as the size of the image set increases. While simple, this experiment demonstrates the utility of Permutation Synchronization for multi-view stereo matching, showing that instead of heuristically propagating local pairwise matches, it can find a much more accurate globally consistent matching at little additional cost.

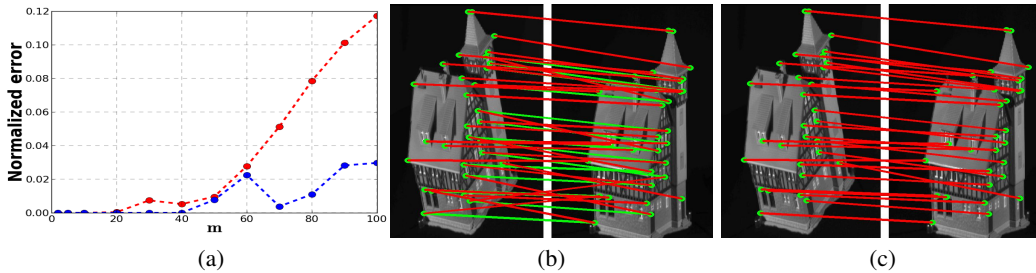

(a)  (b)  (c)

Figure 3: (a) Normalized error as $m$ increases on the House dataset. Permutation Synchronization (blue) vs. the pairwise Kuhn-Munkres baseline (red). (b-c) Matches found for a representative image pair. (Green circles) landmarks, (green lines) ground truth, (red lines) found matches. (b) Pairwise linear assignment, (c) Permutation Synchronization. Note that less visible green is good.

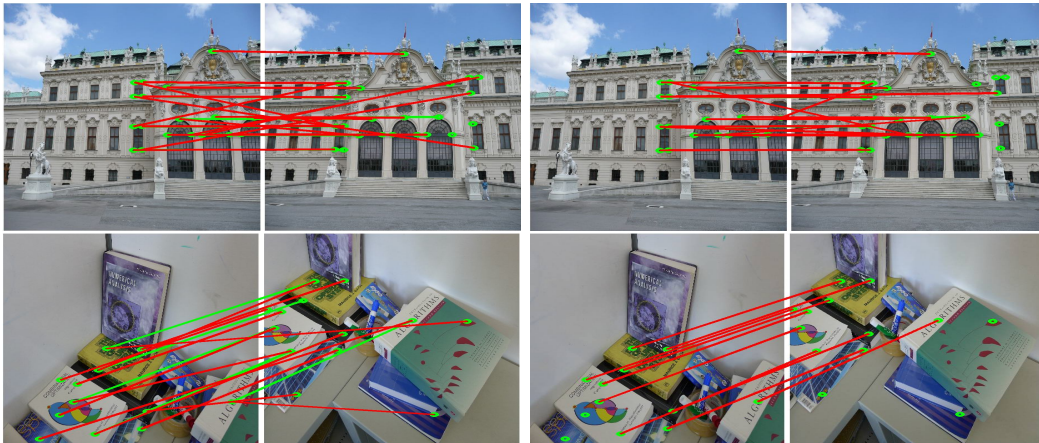

Figure 4: Matches for a representative image pairs from the Building (top) and Books (bottom) datasets. (Green circles) landmark points, (green lines) ground truth matchings, (red lines) found matches. (Left) Pairwise linear assignment, (right) Permutation Synchronization. Note that less visible green is better (right).

**Repetitive Structures.** Next, we considered a dataset with severe geometric ambiguities due to repetitive structures. There is some consensus in the community that even sophisticated features (like SIFT) yield unsatisfactory results in this scenario, and deriving a good initial matching for structure from motion is problematic (see [24] and references therein). Our evaluations included 16 images from the **Building** dataset [24]. We identified 25 "similar looking" landmark points in the scene, and hand annotated them across all images. Many landmarks were occluded due to the camera angle. Qualitative results for pairwise matching and Permutation Synchronization are shown in Fig 4 (top). We highlight two important observations. First, our method resolved geometrical ambiguities by enforcing mutual consistency efficiently. Second, Permutation Synchronization robustly handles occlusion: landmark points that are occluded in one image are seamlessly assigned to null nodes in the other (see the set of unassigned points in the rightmost image in Fig 4 (top)) thanks to evidence derived from the large number of additional images in the dataset. In contrast, pairwise matching struggles with occlusion in the presence of similar looking landmarks (and feature descriptors). For $n = 25$ and $m = 16$, the error from the baseline method (Pairwise Linear Assignment) was $0.74$. Permutation Synchronization decreased this by $10\%$ to $0.64$. The **Books** dataset (Fig 4, bottom) contains $m = 20$ images of multiple books on a "L" shaped study table [24], and suffers geometrical ambiguities similar to the above with severe occlusion. Here we identified $n = 34$ landmark points, many of which were occluded in most images. The error from the baseline method was $0.92$, and Permutation Synchronization decreased this by $22\%$ to $0.70$ (see extended version of the paper).

**Keypoint matching with nominal user supervision.** Our final experiment deals with matching problems where keypoints in each image preserve a common structure. In the literature, this is usually tackled as a graph matching problem, with the keypoints defining the vertices, and their structural relationships being encoded by the edges of the graph. Ideally, one wants to solve the problem for all images at once but most practical solutions operate on image (or graph) pairs. Note

that in terms of difficulty, this problem is quite distinct from those discussed above. In stereo, the *same object* is imaged and what varies from one view to the other is the field of view, scale, or pose. In contrast, in keypoint matching, the background is not controlled and even sophisticated descriptors may go wrong. Recent solutions often leverage supervision to make the problem tractable [25, 26]. Instead of learning parameters [25, 27], we utilize supervision directly to provide the correct matches on a small subset of randomly picked image pairs (e.g., via a crowd-sourced platform like Mechanical Turk). We hope to exploit this 'ground-truth' to significantly boost accuracy via Permutation Synchronization. For our experiments, we used the baseline method output to set up our objective matrix $\mathcal{T}$ but with a fixed "supervision probability", we replaced the $T_{ji}$ block by the correct permutation matrix, and ran Permutation Synchronization. We considered the "Bikes" sub-class from the **Caltech 256 dataset**, which contains multiple images of common objects with varying backdrops, and chose to match images in the "touring bike" class.

Our analysis included 28 out of 110 images in this dataset that were taken "side-on". SUSAN corner detector was used to identify landmarks in each image. Further, we identified 6 interest points in each image that correspond to the frame of the bicycle. We modeled the matching cost for an image pair as the shape distance between interest points in the pair. As before, the baseline was pairwise linear assignment. For a fixed degree of supervision, we randomly selected image pairs for supervision and estimated matchings for the rest of the image pairs. We performed 50 runs for each degree of supervision. Mean error and standard deviation is shown in Fig 5 as supervision increases. Fig 6 demonstrates qualitative results by our method (right) and pairwise linear assignment (left).

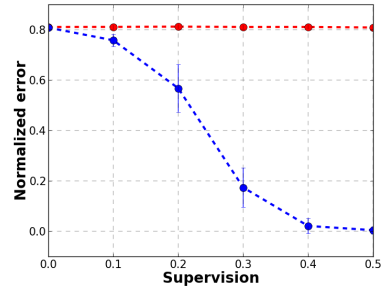

Figure 5: Normalized error as the degree of supervision varies. Baseline method PLA (red) and Permutation Synchronization (blue)

## 5   Conclusions

Estimating the correct matching between two sets from noisy similarity data, such as the visual feature based similarity matrices that arise in computer vision is an error-prone process. However, when we have not just two, but $m$ different sets, the consistency conditions between the $\binom{m}{2}$ pairwise matchings severely constrain the solution. Our eigenvector decomposition based algorithm, Permutation Synchronization, exploits this fact and pools information from all pairwise similarity matrices to jointly estimate a globally consistent array of matchings in a single shot. Theoretical results suggest that this approach is so robust that no matter how high the noise level is, for large enough $m$ the error is almost surely going to be zero. Experimental results confirm that in a range of computer vision tasks from stereo to keypoint matching in dissimilar images, the method does indeed significantly improve performance (especially when $m$ is large, as expected in video), and can get around problems such as occlusion that a pairwise strategy cannot handle. In future work we plan to compare our method to [18] (which was published after the present paper was submitted), as well as investigate using the graph connection Laplacian [28].

**Acknowledgments**

We thank Amit Singer for invaluable comments and for drawing our attention to [18]. This work was supported in part by NSF–1320344 and by funding from the University of Wisconsin Graduate School.

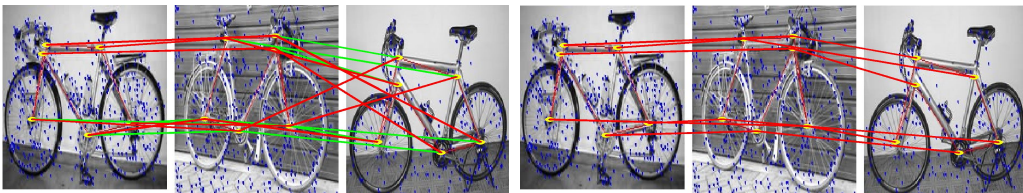

Figure 6:   A representative triplet from the "Touring bike" dataset. (Yellow circle) Interest points in each image. (Green lines) Ground truth matching for image pairs (left-center) and (center-right). (Red lines) Matches for the image pairs: (left) supervision=0.1, (right) supervision=0.5.

## Footnotes

[1] Note that we could equally well have matched the $\sigma_i$'s to any other column of blocks, since they are only defined relative to an arbitrary reference permutation: if, for any fixed $\sigma_0$, each $\sigma_i$ is redefined as $\sigma_i \sigma_0$, the predicted relative permutations $\tau_{ji} = \sigma_j \sigma_0 (\sigma_i \sigma_0)^{-1} = \sigma_j \sigma_i^{-1}$ stay the same.

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
