[Reviews · NeurIPS 2013]

Submitted by Assigned_Reviewer_3

Summary of the paper: This paper provides a permutation synchronization algorithm for matching multiple sets of objects. Most common algorithms use pairwise matching techniques, which can perform badly under noisy data. They provide a novel algorithm using a particular form of distance measure amongst permutations and also theoretically analyse the method under certain noise models. Finally a number of experiments are performed on computer vision applications.

Overall, I liked the paper, and feel that the paper provides a novel algorithm for an important problem particularly in computer vision. The theoretical analysis of the algorithm also seems nice and the experiments seem more than convincing. The writing seems more or less clear, and the work seems to be broadly useful for many application areas, though vision is the main focus of this paper.

A few theoretical issues, which I think, are not sufficiently addressed are the following. In line 154, the authors say that "Solving (6) is computationally difficult because it involves searching a combinatorial space of combination of m permutations". Hence they suggest a relaxation, which they theoretically analyse under certain noise conditions. However, I do not find any NP hardness result stating that this problem is provably NP hard. It is also not true that just having a combinatorial search space implies NP hardness, since there could be efficient polynomial time algorithms exploiting certain structures. So does this problem reduce to a certain NP hard problem?

A second issue is that I do not find sufficient motivation for using the distance measure in equation (4). It might help to motivate this a bit more. Is it primarily, just the ease of the analysis that is the main motivation, or is it that the particular instance of the metric is the best fit? Also how do these compare with the Kendell Tau and other permutation based metrics?

Also it will help to investigate the worst case behaviour of the relaxation. The paper analyses conditions when the relaxation will work, but maybe it might be useful to provide some insights into when these will fail? Maybe some worst case instances, might help to theoretically complete the story (though it is clear that in practice, these examples almost never occur).

Finally a minor issue is that it might help relate this work to other application domains. Since NIPS is mainly a ML conference and this work seems to have a lot of relevance, it might be worth pointing more connections and applications of this. Some application domains which naturally come to mind, are, for instance biology or natural language processing.
Summary: I feel that overall this paper is well written and can have a lot of significance and application to many machine learning problems. It will, however, help to clarify some of the theoretical issues raised above.

Submitted by Assigned_Reviewer_5

The paper's topic is multi-way matching: Suppose you have (noisy, soft) pairwise correspondences between $m$ different sets having $n$ points each. The goal is to find a consistent global assignment between all pairs that is maximally in tune with the local pairwise assignments.

This is an interesting algorithmic problem with important applications. The authors focus on keypoint matching in images from multiple views. Another potential computer vision application could be consistent tracking over time in video.

The authors clearly define notation and formulate multi-way matching as a combinatorial permutation optimization problem. They propose then a spectral relaxation which leads to an eigen-decomposition problem. From the continuous solution they recover the sought after permutations by solving a linear assignment problem. The overall algorithm is fast and scalable.

Positive aspects:
The method presented in the paper seems to be novel as far as I know. The exposition is very clear and the selected application (multi-view point matching) provides a good illustration of the method.

Points that should be improved:

* Section 3 provides some theoretical analysis of the algorithm but it does not pin down a concise mathematical result (in the form of a Proposition).

* The paper does not provide enough information for reproducing the multi-view point matching experiments. It is not clear to me how the key pairwise cost function matrices $T_ji$ are being selected. I suspect that they capture the pointwise similarity between pairs of key-points in the i-th and j-th view, with values between 0 (no similarity) and 1 (full similarity) and normalized such as each row and column of T_ji sums to 1. Is that the case or not?

* The error metric "normalized error" used in experimental evaluation is not properly defined.

* On line 154 it is mentioned that solving the original combinatorial problem (6) is computationally difficult. Is there a proof of that? Do you know whether it is NP-hard? The following book may have some results/pointers that could be useful in that respect:

Burkard et al., "Assignment Problems", SIAM 2008


Some suggestions:

* Is it necessary all point-sets to contain the same number of points $n$? This is an often unnatural requirement. On a related issue, is it possible to explicitly handle outliers, e.g., by adding an extra dummy "outlier" node?

* A standard approach to relaxing combinatorial problems involving permutations is through the Birkhoff polytope. Have you considered that? How is that related to the proposed spectral relaxation?

* It would be interesting to consider online versions of the problem, in which point-sets are added over time. An example is tracking over time key-points in video frames. It would be nice to adapt online eigen-decomposition algorithms to this context.
Summary: The paper contributes a useful algorithm for consistent multi-way matching across point sets. The problem and method are clearly presented and the experimental results are quite convincing. I suggest that the paper gets accepted.

Submitted by Assigned_Reviewer_6

* Summary

This manuscript studies a combinatorial optimization problem that is motivated by a number of applications within the scope of NIPS. It is defined w.r.t. a finite number of finite sets, all of the same cardinality. A feasible solution consists in as many bijections as there are pairs of distinct sets. These bijections are constrained to be consistent in the following sense: For any three sets, A, B, C, if "a" in A is mapped to b in B and b in B is mapped to c in C, then "a" needs to be mapped to c. The objective function is defined w.r.t. so-called "tentative (noisy)" matchings. Instead of solving this combinatorial problem, a relaxation is proposed that can be solved efficiently, by computing leading eigenvectors of a matrix. The solution of the relaxed problem is shown to be robust to noise and also accurate compared to ground truth in a number of practical applications.

* Strengths

- The manuscript addresses a relevant combinatorial optimization problem that has several applications within the scope of NIPS.
- The proposed relaxation of this problem is interesting and original.
- The experimental results suggest that the relaxation may be tight in practice.

* Weaknesses

- A relaxation of an integer program is proposed but not put into perspective. Is it tighter than simpler relaxations? How does it compare to the optimal solution of small problems that can be solved exactly? Is there a guaranteed bound?

- The writing can be improved. In its present form, the objective function is not easy to understand because the notation is convoluted. In order to understand (3), one needs to expand the definitions of d, p, tau, and the inner product. As an alternative, I suggest that the problem should be written as an integer program that is easy to grasp, perhaps in addition to the present formulation.

The readability of Section 2 can be improved. I suggest that the permutations sigma should be defined before the matchings tau and before stating that any set of sigmas induces a set of consistent taus.

The structure of Section 3 can be improved. Statements should be made first informally, then formally, e.g. as lemmata, and then proved.

* Technical questions

- I do not understand the introduction of "tentative (noisy)" matchings in the objective function. After reading the introduction, I would have expected a formulation in terms of weights, positive or negative, one for any possible match (between any two elements of any two distinct sets). Is the objective function (3) a special case of such a general formulation? I think the manuscript can be improved by starting with the general formulation and then specializing this formulation to the problem under consideration.

- Can the combinatorial problem not be solved to optimality by means of a state-of-the-art branch-and-cut solver, at least in applications where the number of elements to be matched is small? If it can be solved to optimality, a comparison of run-times as well as an assessment of the relaxation gap would be interesting.

* Minor remarks

- All equations should be numbered.
Summary: This manuscript proposes an interesting and original relaxation of a relevant combinatorial optimization problem whose solutions are robust against noise and therefore of interest in several applications within the scope of NIPS. Experiments suggest that this relaxation is tight in practice; however, it is not compared with simpler relaxations and exact solutions, e.g. for small problems.
Author Feedback

Author rebuttal: We thank all reviewers for their careful reading of our work. Our response addresses few questions/doubts identified in the review. If accepted, the revised paper will incorporate all other modifications suggested.

R#1, R#2) I do not find any NP hardness result stating that the combinatorial problem is provably NP hard?
Taking the matching likelihood as weights, our synchronization task is a generalization of bipartite matching to many parts or views. The m=3 setting, 3-dim matching, is listed in Garey/Johnson’s Computers and Intractability (Problem SP1 in Appendix A.3.1). It is APX complete.

We are grateful to R#2 for the excellent reference to Burkard et al.

R#1) Sufficient motivation for using the distance measure in (4). Might help to motivate this a bit more. Kendall Tau and other metrics?

Our experiments penalizes all incorrect assignments equally: all points are equally important, which is commonly the case in computer vision applications. Certain settings benefit from other metrics (a preferred ranking of points-to-be-assigned). We think that our model will work for Kendall Tau type measures but we have not worked out the details yet.

R#1) The paper analyses when the relaxation will work, it might be useful to provide some insights into when these will fail?

This is a very interesting comment. The reviewer will see that in simulations, the model works well even under 80% noise. This is nice but not unexpected. The theory (pg. 6, lines 298) provides a sense of such robustness.
Note that the limits of scenarios under which the model is *guaranteed* to recover the true signal are already so severe, that when the assumptions no longer hold, one almost immediately encounters “worst case” instances where the recovery fails completely. In this regime, if the model “accidentally” works, it is simply because of being presented an easy input problem instance.


R#2) Section 3 does not give a concise mathematical result (Proposition).

The Gaussian noise analysis in Section 3 is derived from an elegant result by Benaych-Georges and Nadakuditi. Since our analysis did not involve significant technical steps beyond that paper, we briefly summarized in the text. We are happy to revise and provide a self contained Proposition outlining the key result.

R#2) Does not provide enough information for reproducing the multi-view point matching experiments. … how the key pairwise cost function matrices T are being selected. I suspect that they capture the pointwise similarity between pairs of key-points?

Yes, we use pointwise similarity between keypoints to generate cost function matrices (based on standard functionality in computer vision libraries like OpenCV, VLfeat) followed by Kuhn-Munkres to obtain a stochastic T. We will provide details in the supplement.
Also, we take reproducibility seriously. Our code will be available shortly.

R#2) Is it necessary all point-sets to contain the same number of points $n$? Possible to explicitly handle outliers by adding an extra dummy "outlier" node?
The reviewer is right. As noted in the Experiments, we do not require all point-sets to contain the same number of points which will clearly be a serious restriction. In fact, the stereo matching and other experiments indeed add dummy nodes, exactly as the reviewer suggests. In the end, we automatically assign a dummy node to outliers (see Figure 4).

R#3) A relaxation of an IP is proposed. Tighter than simpler relaxations? How does it compare to the optimal solution of small problems that can be solved exactly?

We are not sure we fully understand this concern, so a short clarification is needed.

Our goal is not to argue that the spectral relaxation is “tighter” than alternative formulations. Instead, we show that for a rather broad and permissive range of values for n and m (also noise behavior), the solution from solving the eigen decomposition is EXACT.

R#3) Understand the introduction of "tentative (noisy)" matchings in the objective function. I would have expected a formulation in terms of weights. Is the objective function (3) a special case?
“Tentative” was used to refer to noisy matching matrices that might produce incorrect matches between pair of objects, if matched locally.
The optimization problem that we solve (see eq (5)) allows assigning general real valued weights on each match. Objective function (3) is a special case of (5) (written in terms of the permutation group for brevity).

R#3) Can the combinatorial problem not be solved to optimality by a state-of-the-art solver...where n is small? A comparison of run-times would be interesting.

We solve the problem to optimality. The only error in our plots comes from introducing various levels of noise.
The reviewer will likely agree that the number of constraints and the number of variables to specify an IP for the types of n and m values used in our experiments, grows very rapidly. Beyond the smallest problem instances, Branch/Bound and Branch/Cut are really not viable alternatives here because number of nodes visited can be exponential in n.